# Ruxolitinib as Adjunctive Therapy for Hemophagocytic LymPhohistiocytosis after Liver Transplantation: A Case Report and Literature Review

**DOI:** 10.3390/jcm11216308

**Published:** 2022-10-26

**Authors:** Kang He, Shanshan Xu, Lijing Shen, Xiaosong Chen, Qiang Xia, Yongbing Qian

**Affiliations:** 1Department of Liver Surgery, Renji Hospital, School of Medicine, Shanghai Jiao Tong University, Shanghai 200127, China; 2Shanghai Engineering Research Center of Transplantation and Immunology, Shanghai 200127, China; 3Shanghai Institute of Transplantation, Shanghai 200127, China; 4Department of Hematology, Renji Hospital, School of Medicine, Shanghai Jiao Tong University, Shanghai 200127, China

**Keywords:** ruxolitinib, hemophagocytic lymphohistiocytosis, liver transplantation

## Abstract

Hemophagocytic lymphohistiocytosis (HLH) is a rare but potentially fatal hyperinflammatory disorder characterized by dysfunctional cytotoxic T and natural killer cells. Liver transplantation is a predisposing factor for HLH. High mortality rates were reported in 40 cases of HLH following liver transplantation in adults and children. Herein, we describe a case of adult HLH triggered by cytomegalovirus (CMV) infection shortly after liver transplantation. The patient was successfully treated with ruxolitinib combined with a modified HLH-2004 treatment strategy. Our case is the first to report the successful use of ruxolitinib with a modified HLH-2004 strategy to treat HLH in a solid organ transplantation recipient.

## 1. Introduction

Hemophagocytic lymphohistiocytosis (HLH), also known as hemophagocytic syndrome (HPS), is a rare but life-threatening disorder characterized by a high-grade fever, splenomegaly, cytopenia, and hyperferritinemia. It is believed that NK cell dysfunction, activated cytotoxic T-cell activity, and the cytokine storm constitute the underlying mechanism of HLH [1]. HLH can be divided into primary HLH and secondary HLH according to its cause. Primary HLH is considered to be genetic and often occurs in children with an underlying genetic defect. Bone marrow transplant is the only radical treatment for primary HLH [2]. In contrast, secondary HLH can occur at any age and is associated with infection, malignancy, autoimmune disease, transplantation, and drugs [1]. These factors may act as triggers or predisposing factors to the disease, or both. Viral infection is the most frequent trigger, and about 6% of cases are triggered by a cytomegalovirus (CMV) [1]. HLH is a rare disease, but has a very high mortality rate of around 41%, probably due to the nonspecific manifestations and delayed diagnosis [1].

After liver transplantation (LT), patients receive immunosuppression treatment and as a result are susceptible to infections and immune dysregulation. HLH after LT is often overlooked as a fever and cytopenia both of which are common after a liver transplant, and the initial manifestations of HLH are also similar to other complications, such as posttransplant lymphoproliferative disorders (PTLD), graft-versus-host disease (GVHD), autoimmune hemolytic anemia (AIHA), and sepsis [1,3]. To date, 21 pediatric cases (age ≤ 18 years) and 19 adult cases of HLH following liver transplantation (LT) have been reported in the literature, with a very high mortality rate of 55% (22/40) (Table 1). The high mortality rate may be attributed to a delayed diagnosis and a debatable treatment. The widely used HLH-2004 treatment protocol, including steroids (dexamethasone), immunosuppressive agents (cyclosporine A), and chemotherapy drugs (etoposide), was first proposed for pediatric patients with primary HLH. However, this protocol may not be appropriate for LT patients as they are already immunosuppressed [4].

Ruxolitinib is an inhibitor of JAK1/2 (the Janus family kinase 1 and 2) and can inhibit the release of cytokines. It is mainly used in the hematological field and also in COVID-19 patients [5]. There are case reports and pilot studies using ruxolitinib as a salvage therapy or first-line treatment for HLH patients showing good efficacy and safety [6,7,8,9,10,11,12,13,14,15]. However, it has never been used for post-transplant patients with HLH. In this study, we present one case of adult CMV-associated HLH following liver transplantation successfully treated by ruxolitinib and a modified HLH-2004 protocol. This report provides an alternative method for the treatment of HLH following LT.

**Table 1 jcm-11-06308-t001:** Cases of HLH after liver transplantation.

No	Author	Age (Years)	Dx	Donor Type	Time Post LT	IS regimen	Triggers	Treatment	Outcome
1	Chisuwa [16]	0.75	Biliary atresia	LRLT	15 d	Steroid, FK506	Unx	Steroid, IVIg, G-CSF, stopped IS	Died
2	Chisuwa [16]	1	Biliary atresia	LRLT	134 d	Steroid, FK506	Unx	G-CSF, stopped IS	Died
3	Karasu [17]	38	HBV/HDV	LRLT	131 d	Steroid, FK506	Unx	IVIg, G-CSF, PE, stopped IS	Alive
4	Lladó [18]	63	AIH	DDLT	75 d	CsA, basiliximab	Unx	Steroid, IVIg	Died
5	George [19]	10	Unx	Unx	6 y	FK506	EBV	Steroid, Stopped IS, etoposide	Alive
6	Tania [20]	37	Unx	LDLT	11 d	FK506	Unx	Steroid, IvIg, PE, CHDF	Died
7	Hardikar [21]	2.2	Unx	DDLT	15 d	Steroid, FK506, Aza	CMV	Stop IS, G-CSF, IVIg	Alive
8	Akamatsu [22]	59	HCV	LDLT	138 d	FK506	Aspergillus	Supportive	Died
9	Akamatsu [22]	49	PBC	LDLT	315 d	Steroid, FK506	CMV	Steroid, IVIg, PE, FK506 to CsA, Sx	Died
10	Akamatsu [22]	48	HCV	LDLT	50 d	Steroid, FK506, OKT3	HCV	G-CSF	Died
11	Yoshizumi [23]	63	HBV	LDLT	13 d		SFSS	Steroid, change IS, IVIg, G-CSF	Alive
12	Dharancy [24] L/K	49	Polycystic liver kidney disease	Combined Liver-kidney (CLK)	19 d	Steroid, CsA, MMF, basiliximab	HHV6	Supportive	Died
13	Zhang [25]	6	Unx	LRLT	8 d	Steroid, FK506	EBV	Steroid, IVIg, G-CSF, stopped IS, etoposide	Alive
14	Satapathy [26]	25	AIH	DDLT	6 m	Steroid, FK506, MMF	Still’s disease	Steroid, anakinra	Alive
15	Somyama [27]	57	HCV/HCC	LDLT	32 d	Steroid, FK506	CMV/HCC	Steroid, IVIg, G-CSF	Died
16	Somyama [27]	63	HBV/HCC	LRLT	81 d	Steroid, FK506	Unx	Steroid, IVIg, G-CSF	Died
17	Rodríguez-Medina [28]	63	HCV	Unx	4 y	Steroid, CsA, Aza	TB	G-CSF	Died
18	Jha [29] 2015	2	Extrahepatic biliary atresia	LRLT	9 m		EBV	Reduced IS; rituximab	Alive
19	Vijgen [30]	66	Alcohol-related liver disease, HCC	DDLT	1 y	FK506, MMF	HHV8	Steroid, stopped IS, rituximab	Died
20	Iseda [31]	63	HBV	LDLT	12 d	Steroid, FK506, MMF	SFSS	Steroid, IVIg, G-CSF, stopped MMF, reduced FK506	Died
21	Cohen [32]	31	Drug-induced liver disease	DDLT	4 y	FK506	HHV8	Steroid, IVIg, rituximab, reduced FK506	Died
22	Jarchin [33]	18	Neonatal hepatitis	LRLT	17 y	FK506	Unx	Steroid, etoposide, stopped FK506	Died
23	Valenzuela [34]	48	AIH, PBC	DDLT	13 d	Steroid, FK506	EBV	Steroid, etoposide, IS switched to CsA	Died
24	Yamada [35]	2	Biliary atresia	Liver inteatinal transplantation	254 d	Steroid, FK506, everolimus	EBV	Steroid, etoposide, reduced IS, THP-COP	Died
25	Chesner [36]	58	AIH	Unx	16 y	Steroid, FK506, Aza	EBV	supportive	Died
26	Arikan [37]	1	Caroli disease	Unx	6 m	Steroid, FK506, sirolimus, MMF	CMV	Steroid, IVIg, rituximab	Died
27	Arikan [37]	2	Biliary atresia		8 m	Steroid, FK506, sirolimus	EBV	Steroid, IVIg, VP16, rituximab	Alive
28	Arikan [37]	1.5	Unx cholestasis		7 m	Steroid, FK506	HHV8	Steroid, IVIg, VP16	Alive
29	Arikan [37]	9	PFIC		4 m	Steroid, FK506, sirolimus	Acinetobacter	IVIg	Died
30	Arikan [37]	0.5	Biliary atresia		10 m	FK506	EBV	IVIg, VP16, rituximab	Alive
31	Arikan [37]	0.5	Unx cholestasis		7 m	Steroid, FK506	Klebsiella	Steroid, IVIg, VP16	Alive
32	Arikan [37]	0.67	Biliary atresia		9 m	Steroid, FK506	EBV	Steroid, IVIg, VP16, rituximab	Alive
33	Arikan [37]	0.67	Unx		5 m	Steroid, FK506, MMF	CMV, EBV	Steroid, IVIg, rituximab	Died
34	Arikan [37]	0.75	Unx cholestasis		6 m	Steroid, FK506	EBV	Steroid, IVIg, rituximab	Alive
35	Arikan [37]	2.1	PFIC		10 m	FK506	EBV	IVIg, VP16, rituximab	Alive
36	Arikan [37]	0.4	Unx cholestasis		8 m	FK506	EBV	Steroid, IVIg, rituximab	Alive
37	Arikan [37]	1	Bile salt synthesis defect		10 m	FK506, sirolimus	EBV	IVIg, rituximab	Alive
38	Gandotra [38]	47	HBV/HCV/HCC	LDLT	3 y	Steroid	TB	Steroid, IVIg	Alive
39	Nakajima [39]	2	EBV-LPD	LRLT	10 d		EBV	Steroid, etoposide, HCT	Alive
40	Ferreira [40]	52	Cryptogenic cirrhosis	DDLT	2 m	Steroid, FK506, MMF	TB	Steroid, etoposide, stopped FK506	Died

AIH: autoimmune hepatitis, Aza: azathioprine, CMV: cytomegalovirus, CsA: cyclosporin A, d: days, DDLT: deceased donor liver transplantation, EBV: Epstein-Barr virus, EBV-G-CSF: granulocyte colony-stimulating factor, LPD: EBV associated T and NK cell lymphoproliferative disease, FK506: tacrolimus, HBV: hepatitis B virus, HCV: hepatitis C virus, HDV: hepatitis D virus, HCC: hepatocellular carcinoma, HCT: hematopoietic cell transplantation, HHV: human herpes virus, IVIg: intravenous immunoglobin, IS: immunosuppression, LDLT: living donor liver transplantation, LRLT: living related donor liver transplantation, m: months, MMF: mycophenolate mofetil, PBC: primary biliary cholangitis, PE: plasma exchange, PFIC: progressive familial intrahepatic cholestasis, SFSS: small-for-size syndrome, Sx: splenectomy, TB: tuberculosis, Unx: unknown, VP-16: etoposide, y: years.

## 2. Case Report

A 53-year-old male presented with fulminant hepatic failure, probable drug-induced liver injury and alcohol-induced hepatitis. Serologic tests for hepatitis A-D, Epstein-Barr virus (EBV), and tuberculosis were all negative. The autoimmune workup was normal. The CMV DNA was detected by PCR in peripheral blood with 4.64$\;\times \;10^{4}\;$copies/mL, with a positive cytomegalovirus (CMV) IgG and a negative IgM.

On admission, the patient’s vitals were within normal ranges and his temperature was 36.5 °C. Initial examination showed normal hematological findings and an abnormal liver function and coagulopathy: total bilirubin 544.6 μmol/L, direct bilirubin 386.8 μmol/L, alanine aminotransferase (ALT) 537 U/L, aspartate aminotransferase (AST) 373 U/L, PT 24.1 s, INR 2.3, and fibrinogen 1.15 g/L.

On day 10 of admission, the patient underwent a deceased donor liver transplantation. The donor was a 50-year-old male with a height of 165 cm and weight of 55 kg (BMI 20.2 kg/m^2^), diagnosed with brain death induced by intracranial hemorrhage. The donor was CMV-seronegative and had an identical blood type (type B) to the patient. The graft weight was 1515 g. The liver transplant took 6 h with 35 min of anhepatic time. During the surgery, 500 mg methylprednisolone was used for immunosuppression induction. The patient received a standard immunosuppression treatment with methylprednisolone tapering (tapering from 240 mg to 20 mg within one week), mycophenolate mofetil (720 mg/day), tacrolimus (4 mg/day), and basiliximab (20 mg on POD 0 and 4) after the surgery.

The post-transplant outcome is shown in Figure 1. On postoperative day (POD) 3, the patient developed a high-grade fever (above 38.5 °C), thrombocytopenia and anemia: WBC 4.97$\;\times \;10^{9}$/L (neutrophil 4.36$\;\times \;10^{9}$/L), hemoglobin 73 g/L, and platelets 43$\;\times \;10^{9}$/L. Experimental anti-infection treatment was initiated with vancomycin and meropenem to treat the fever. Granulocyte colony-stimulating factor (G-CSF, 75 μg/day) was administered three times on POD 5, 9, and 17 to prevent white blood cell (WBC) count decline, but only temporary improvement was observed. A blood culture was immediately performed and repeated twice, revealing no bacterial infections. Next Generation Sequencing screening, bacterial culture of the intubation and drainage liquid, G test, GM test, TORCH test, respiratory virus test, and a chest X-ray were also all negative. On POD 8, the CMV DNA in peripheral blood increased to 4.64$\;\times \;10^{6}$ copies/mL, and ganciclovir (400 mg/day) was initiated to treat the infection. Considering the normal hepatobiliary enzymes, bilirubin levels and normal daily ultrasonogram, immunosuppression therapy (tacrolimus, mycophenolate mofetil and methylprednisolone) was temporarily stopped in order to promote an immune response to overcome the infection.

Despite the use of ganciclovir, the patient’s inflammatory factors, fever and pancytopenia showed no improvement. On POD 22, the WBC count declined to 1.21$\;\times \;10^{9}$/L (neutrophil 1.09$\;\times \;10^{9}$/L), and the patient exhibited a persistent high-grade fever. He was then tested for HCMV drug-resistant mutation, but the result was negative. Therefore, hemophagocytic lymphohistiocytosis (HLH) was strongly suspected even without a relevant family history. Bone marrow aspiration was performed on POD 23 and revealed hemophagocytosis (Figure 2). Laboratory findings: hemoglobin 58 g/L, platelets 72$\;\times \;10^{9}$/L, WBC 1.65 × 10^9^/L (neutrophils 1.54$\;\times \;10^{9}$/L), ferritin 1592.5 ng/mL, fasting triglyceride 1.06 mmol/L, fibrinogen 2.78 g/L, sIL-2R 6150 U/mL, and low NK cell activity (2.1%). The patient met seven out of the eight HLH-2004 diagnostic criteria [4] and was diagnosed with HLH following LT. According to the HLH-2004 protocol, a combination of intravenous immunoglobulin (25 g/day), G-CSF (150 μg/d), steroids (methylprednisolone pulse 80 mg), and etoposide (100 mg/d) was immediately administered [4]. The JAK1/2 inhibitor ruxolitinib is a novel treatment for HLH and was started the next day after diagnosis in combination with dexamethasone. The dose of ruxolitinib was maintained at 5 mg/day until discharge, and dexamethasone was tapered from 20 mg/day to 5 mg/day over three weeks and maintained at 5 mg/day until discharge (POD 28 15 mg/day, POD 34 10 mg/day, POD38 7.5 mg/day, and POD 44 5 mg/day). The patient’s fever improved rapidly following treatment initiation. On POD 25, the patient’s temperature returned to normal. The inflammatory factors IL-6 and C-reactive protein declined rapidly after the treatment. Ferritin and sIL-2R are two reliable prognostic factors of HLH; sIL-2R showed a significant decline while ferritin showed a gradual decrease after treatment initiation [41]. The WBC, hemoglobin and platelet levels showed improvement after two weeks of HLH and supportive treatment. GSF and recombinant human interleukin-11 were given to improve the platelet level, and a therapeutic plasma exchange was performed to clear the excessive cytokines. On POD29, tacrolimus was resumed due to a slight increase in liver enzymes. On POD33, the patient tested positive for carbapenem-sensitive klebsiella pneumoniae (CSKP). Minocycline and meropenem were used. Dexamethasone was reduced to 10 mg/day the next day and then to 7.5 mg/day on POD38, which led to a significant improvement in neutropenia.

With a stable temperature and lab findings, the patient was discharged on POD 50. His maintenance treatment included ruxolitinib (5 mg/day), dexamethasone (4.5 mg/day), tacrolimus (1 mg/day), ganciclovir, and entecavir. Regular follow-ups were performed after discharge. Dexamethasone was reduced by one tablet (0.75 mg) per week forone1 month, which resulted in a rapid decline in WBC count, which was successfully treated by increasing ruxolitinib to 10 mg/day. On POD 71 (three weeks after discharge), the peripheral blood CMV DNA was undetectable, and ganciclovir was stopped. Three months after discharge, dexamethasone (2.25 mg/day) was changed to prednisone (5 mg/day), and ruxolitinib was stopped. The patient’s temperature remained stable and the patient is doing well to date (12 months).

## 3. Discussion

HLH is hard to diagnose due to its non-specific symptoms. The patient’s CMV DNA was 4.64$\;\times \;10^{4}$ copies/mL before transplant. After liver transplantation, with the use of immunosuppression agents, the CMV DNA increased to 4.64$\;\times \;10^{6}$ copies/mL on POD8. The initial symptoms were thought to be caused by CMV reactivation, and differential diagnoses such as HLH were only considered after failure of the initial treatment. Failure to treat CMV with ganciclovir in a timely manner was indeed one of the possible causes of secondary HLH in our case. Basiliximab, plus a delayed and low-dose immunosuppressive regimen of tacrolimus were intended to reduce further CMV flareups after the transplant. Bone marrow aspiration was performed to confirm the diagnosis, which is not a mandatory diagnostic criterion, but provides valuable information for differential diagnosis.

The HLH-2004 treatment protocol includes a combination of chemotherapy (etoposide), immunosuppressive therapy (cyclosporin A, CsA), glucocorticoid, and bone marrow transplantation, which has a 5-year survival rate of about 61% [4]. CsA is proven to be beneficial in pediatric and autoimmune cases and is recommended in the HLH-2004 protocol, while tacrolimus’s efficacy has not been proved [1]. Hence, some clinicians choose to change the immunosuppression to CsA. It has been reported that tacrolimus has a stronger inhibitory effect on CD8 T cells and has shown efficacy in the treatment of HLH [42,43]. Thus, the need for switching tacrolimus to CsA is controversial (Table 1).

However, the HLH-2004 treatment protocol is a conflicting strategy for patients with HLH after transplantation, as these patients are already in an immunosuppressive state, which could also be a predisposing factor for HLH. Clinicians argued that CsA was ineffective in preventing HLH progression after solid organ transplantation [44]. The myelotoxicity, hepatotoxicity, and nephrotoxicity of etoposide may also have resulted in complications. Therefore, the immunosuppressants were stopped, and etoposide was given only three times during the early stages. Most importantly, the novel drug ruxolitinib was added to the treatment strategy.

Ruxolitinib is an oral JAK1/2 inhibitor and can inhibit the production of interferon-γ (IFN-γ) [6]. It has been approved to treat myelofibrosis, polycythemia, and steroid-refractory acute graft-versus-host disease (SR-aGVHD) by the US Food and Drug administration [45]. Considering the complexity of the cytokine storm in HLH, the pathogenesis is likely mediated by a network of cytokines rather than a single one. However, it is generally believed that the pathway involving STAT1 (signal transducer and activator of transcription 1), JAK1, JAK2, and IFN-γ is the common pathway of HLH [46,47,48]. Ruxolitinib has been reported to be effective in treating 61 secondary HLH patients, including 19 adults and 42 children. In nine cases, ruxolitinib was used as the first-line treatment, while in other cases it was used as a salvage therapy. However, ruxolitinib has never been reported to be used in HLH after solid organ transplantation.

In 2019, Ahmed et al. conducted a small pilot study that included seven patients, which had shown that ruxolitinib is an active and well-tolerated treatment for secondary HLH in adult patients and can be managed through an outpatient clinic [6]. Two of the patients’ conditions were triggered by an autoimmune disorder, and three were idiopathic [6]. Among the seven patients, three of them achieved a complete response, and the others achieved a partial response with improved cytopenia within the first week [6]. The use of ruxolitinib allowed the reduction or discontinuation of corticosteroids [6]. All the patients were alive with a median follow-up of 490 days (range: 190-1075 days) [6]. However, one patient was diagnosed with primary HLH and suffered a relapse, while another patient experienced a drug-related adverse event (neuropathic foot pain) and stopped the treatment [6]. The only serious side effect of ruxolitinib was grade 4 febrile neutropenia without any drug-related death [6]. Currently, a multi-center phase 2 clinical trial is investigating the efficacy of ruxolitinib (NCT04551131). In addition, participants are being recruited for a phase 4 clinical trial focusing on malignancy-related HLH and the effect of ruxolitinib (NCT04999878).

Wang et al. conducted a clinical trial, which included 34 children with refractory/relapsed HLH [7]. Ruxolitinib was given as a salvage therapy after the HLH-94 regimen failure. Twenty five children responded to the therapy and five of them had complete remission. Leukopenia and thrombocytopenia were common side effects, but all the patients could tolerate them. At the end of the study, 15 patients died, and the median survival time was 22 weeks. The clinicians argued that ruxolitinib could alleviate the hyperinflammatory state but was not sufficiently potent. They suggested combining ruxolitinib with other treatment strategies to achieve better outcomes. A large-scale clinical trial is being conducted to validate the effectiveness of the DEP-ruxolitinib regimen (NCT03533790)

In 2019, Meng performed a single-center retrospective analysis evaluating 12 patients who developed SR-aGVHD after allogeneic hematopoietic stem cell transplantation (allo-HSCT), complicated by Epstein-Barr virus-associated HLH [8]. Among the 12 patients, seven achieved a complete response, three achieved a partial response, and two exhibited treatment failure. The side effects only included neutropenia (grade 3 to 4) and thrombocytopenia (grade 3 to 4). Ruxolitinib was apparently effective for patients with EBV-HLH combined with SR-aGVHD.

In 2020, Wei conducted a retrospective analysis of pediatric cases with refractory or recurrent HLH and found that ruxolitinib was effective as a salvage therapy [9]. Among the nine analyzed cases, five cases of HLH were secondary to EBV, one was secondary to an autoimmune disorder, two were familial HLH, and one was idiopathic. Before the use of ruxolitinib, all of them were treated according to the HLH-94 treatment strategy but did not show improvement. After receiving a continuous 28-day course of ruxolitinib, three of them achieved partial remission, five improved and one died.

Furthermore, six case reports argued that ruxolitinib is beneficial to refractory HLH both for children and adults. Slostad reported a case of histoplasmosis-related HLH treated with ruxolitinib as a first-line treatment, which achieved a rapid improvement [10]. Zandvakili presented a case of autoimmune disease-related HLH, also using ruxolitinib as a first-line therapy with a good outcome even after the patient developed sepsis [11]. Goldsmith reported two HLH cases, one autoimmune disorder-associated and the other EBV-related [12]. Ruxolitinib was used as a salvage therapy and led to a rapid symptom improvement. Zhao reported a pediatric case of EBV-associated HLH successfully treated with ruxolitinib as a bridge to allogeneic stem cell transplantation [13]. Brogile reported a pediatric case of idiopathic HLH receiving ruxolitinib after several failed treatments, including the biological treatment anakinra. The patient’s fever subsided within 24 h of initiating ruxolitinib, and recovery was achieved without other treatments [14]. Ono successfully used ruxolitinib in a pediatric patient who developed HLH after hematopoietic cell transplantation [15].

The most commonly reported adverse effects of ruxolitinib were hematologic adverse reactions including thrombocytopenia and anemia [49,50]. Dizziness, headache, and infections are common nonhematologic adverse effects [45,49]. Hepatotoxicity is rare and only moderate liver toxicity was reported [51,52]. For patients with hepatic impairment, the dosage of ruxolitinib should be modified according to platelet counts [45]. Both tacrolimus and ruxolitinib are mainly metabolized through cytochrome P450 (CYP) 3A4 enzyme, but no drug interaction between them has been reported [53,54]. For ruxolitinib, dose reduction is only needed when a potent CYP3A4 inhibitor is concomitantly used [53]. However, tacrolimus has a narrow therapeutic index and is more likely to be influenced by other drugs that also metabolize through CYP3A4 [54]. So close monitoring of tacrolimus whole blood concentrations is recommended. Based on this information, we decided to lower the dose of ruxolitinib to 5 mg/day, in comparison to 20 mg twice daily for myelofibrosis patients [45]. The platelet count and liver function were closely monitored, and tacrolimus trough concentration was tested every day.

Our patient developed a high-grade fever shortly after the operation and only CMV infection was detected. However, no symptom improvement was observed following the initiation of ganciclovir. Furthermore, steroids and immunosuppression drugs, as the two major components of HLH-2004 treatment, had already been given for around two weeks. Hence, we considered that the patient was unresponsive to the HLH-2004 treatment protocol. In addition, a pilot study reported ruxolitinib as a first-line treatment for adult secondary HLH [6], and case reports described the use of ruxolitinib as a first-line therapy to treat Hematopoietic Cell Transplantation-associated HLH [10,11,15]. Moreover, some clinicians argued that ruxolitinib may exert a synergistic effect with HLH-2004 treatment [55]. Ruxolitinib was added to our treatment strategy and the dosage and frequency of etoposide were significantly reduced (100 mg three times instead of 150 mg ten times). Dexamethasone was tapered much more rapidly and was maintained at a low level (within three weeks instead of eight weeks according to HLH-2004 strategy).

Our case is the first to report the use of ruxolitinib as an initial therapy instead of a salvage therapy in HLH patients after solid organ transplantation. The patient’s fever subsided within 24 h, with a rapid improvement in ferritin, and sIL-2R levels. However, ruxolitinib treatment resulted in a slow improvement in cytopenia, possibly due to the use of ganciclovir and tacrolimus or the adverse effect of ruxolitinib (neutropenia and thrombocytopenia). The neutropenia and thrombocytopenia spontaneously improved after a reduction in the dexamethasone dosage. The patient is now doing fine with a normal liver function and minimal immunosuppressants almost one year after liver transplantation.

In summary, we present an adult case of secondary HLH associated with CMV infection following a liver transplant successfully treated with a modified HLH-2004 treatment protocol, which included ruxolitinib. The rapid improvement and good prognosis indicate that ruxolitinib may be useful as an initial therapy for patients with HLH after solid organ transplantation.

## Figures and Tables

**Figure 1 jcm-11-06308-f001:**
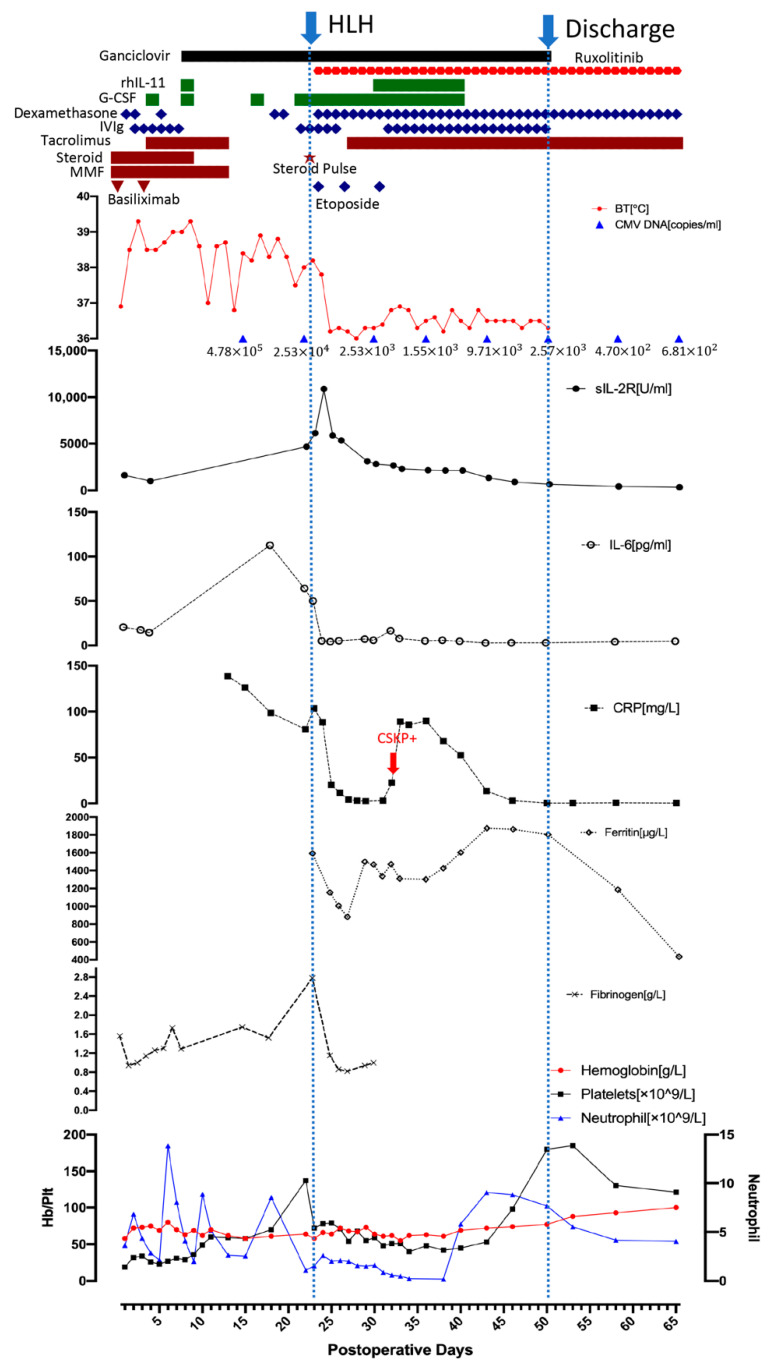
The posttransplant outcome of the patient. (BT: body temperature, CRP: C-reactive protein, CSKP: carbapenem-sensitive klebsiella pneumoniae, G-CSF: granulocyte colony-stimulating factor, Hb: hemoglobin, IL-6: interleukin-6, IVIg: intravenous immunoglobulin, MMF: mycophenolate mofetil, Plt: platelet, rhIL-11: recombinant human interleukin-11, sIL-2R: soluble interleukin-2 receptor).

**Figure 2 jcm-11-06308-f002:**
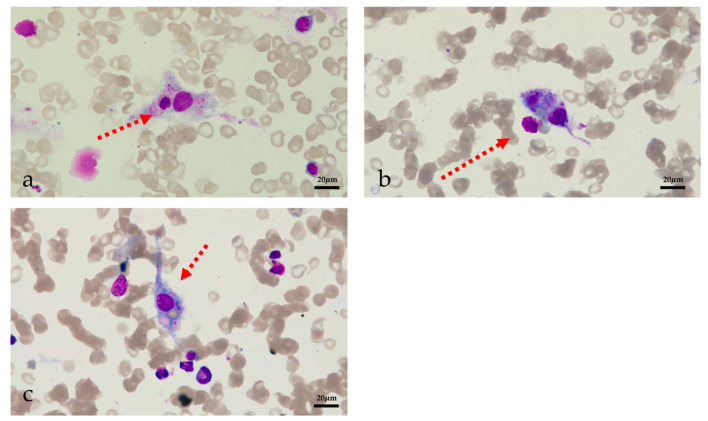
Image of bone marrow aspiration (**a**) phagocytic neutrophils (**b**) phagocytic platelets (**c**) erythrophagocytosis.

## Data Availability

Not applicable.

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
