# Peer review of "Ruxolitinib as Adjunctive Therapy for Hemophagocytic LymPhohistiocytosis after Liver Transplantation: A Case Report and Literature Review"

_jcm, 2022, doi:10.3390/jcm11216308_

Round 1

Reviewer 1 Report

Major

This report is informative for HLH treatment after liver transplantation. For this reason, please inform more about drug interaction between Tacrolimus and Ruxolitinib and adverse effect of Ruxolitinib, especially hepatotoxicity.

Minor

Figure 1 needs unit for the y-axis and ‘abbreviations’ for IVIG, MMF, CSKP+, rhIL-11, ect..

P1 L35, cytomegalovirus. à cytomegalovirus (CMV).

P2 L58, cytomegalovirus (CMV) à CMV

The following sentence is inappropriate as scientific journal.

(1)   P2 L68, Why don’t you suggest operation and ischemic time? “The surgical procedure was uneventful and rapidly completed.”

(2)   P2 L87, Why don’t you change the verb from “fight” to “overcome?

“~to promote immune response to fight the infection.”

Author Response

Reviewer 1

Major

This report is informative for HLH treatment after liver transplantation. For this reason, please inform more about drug interaction between Tacrolimus and Ruxolitinib and adverse effect of Ruxolitinib, especially hepatotoxicity. 

We added information about adverse effect and drug interaction in the discussion section (Line 264-280).

Minor 

Figure 1 needs unit for the y-axis and ‘abbreviations’ for IVIG, MMF, CSKP+, rhIL-11, ect.. 

The missing units were added and the abbreviations were explained.

P1 L35, cytomegalovirus. à cytomegalovirus (CMV).

P2 L58, cytomegalovirus (CMV) à CMV

Line 35 and 58 (now Line 35 and 63) were all changed according to the reviewer’s advice.

The following sentence is inappropriate as scientific journal. 

(1)   P2 L68, Why don’t you suggest operation and ischemic time? “The surgical procedure was uneventful and rapidly completed.”

L68, now L74 was changed accordingly.

(2)   P2 L87, Why don’t you change the verb from “fight” to “overcome?

“~to promote immune response to fight the infection.”

L87, now L94 was changed accordingly.

Reviewer 2 Report

The authors present an interesting case of HLH acquired from CMV infection following liver transplant. The authors do a good job explaining the timeline of events and discuss relevant literature. I have the following questions/ comments :

- The patient had CMV viremia at the time of transplant and post-operatively, the patient was not started on empiric Ganciclovir till POD-8 when the CMV counts were significantly higher (increased by a log of 2).

- At the same time, the recipient was started on full immunosuppression regimen knowing that the already had CMV viremia. The recipient was also given Basiliximab which probably contributed to the overimmunosuppression of this recipient with further elevation of CMV counts. The authors should discuss this.

- As for the efficacy of Ruxolitinib, the authors used it in combination with a lot of other modalities including IVIg, Dexamathasone, etoposide etc. I am unsure if the improvement in the recipient was from Ruxolitinib or the other treatments.

- Authors must have checked CMV PCRs during the course of this prolonged hospitalization and after discharge. They should mention the counts here in the case report to see the correlation of CMV counts with the symptoms.

- In other cases cited and reviewed here, some other authors have transitioned patients from Tacrolimus to Cyclosporine. What do the authors think about that as part of the treatment strategy?

Author Response

Reviewer2
The authors present an interesting case of HLH acquired from CMV infection following liver transplant. The authors do a good job explaining the timeline of events and discuss relevant literature. I have the following questions/ comments :

- The patient had CMV viremia at the time of transplant and post-operatively, the patient was not started on empiric Ganciclovir till POD-8 when the CMV counts were significantly higher (increased by a log of 2).

- At the same time, the recipient was started on full immunosuppression regimen knowing that the already had CMV viremia. The recipient was also given Basiliximab which probably contributed to the overimmunosuppression of this recipient with further elevation of CMV counts. The authors should discuss this.

Failure to treat CMV infection with ganciclovir in a timely manner is indeed one of the possible causes of secondary HLH in our case. Basiliximab plus a delayed and low-dose immunosuppressive regimen of tacrolimus is intended to reduce further CMV flareup after surgery. We have added this part in our discussion part.

- As for the efficacy of Ruxolitinib, the authors used it in combination with a lot of other modalities including IVIg, Dexamathasone, etoposide etc. I am unsure if the improvement in the recipient was from Ruxolitinib or the other treatments.

For ruxolitinib efficacy, we acknowledge that one case report cannot prove it. However, as we stated in the discussion section, ruxolitinib allowed us to reduce etoposide and taper dexamethasone faster. Also, the treatment for HLH always include a combination of different drugs, like HLH-2004 treatment including steroids, chemotherapy and so on.

- Authors must have checked CMV PCRs during the course of this prolonged hospitalization and after discharge. They should mention the counts here in the case report to see the correlation of CMV counts with the symptoms.

Thank you for your kind remind. The CMV DNA counts was added in Figure 1. The posttransplant outcome of the patient.

- In other cases cited and reviewed here, some other authors have transitioned patients from Tacrolimus to Cyclosporine. What do the authors think about that as part of the treatment strategy?

We explained the reasons why some authors change tacrolimus to cyclosporine in discussion section (Line 181-186).

Reviewer 3 Report

The authors present a very well-described therapy for HLH - LT. they only must make some minor modifications as:

In the introduction, it would be adequate to describe the mechanism of ruxolitinib and why the authors chose it! (briefly)

Line 46: Briefly report the most common drugs in the HLH-2004 treatment protocol.
Figure 2: bar scale is missing, please put it on the images.
Line 172-178, the authors must inform the reference

Author Response

Reviewer3

The authors present a very well-described therapy for HLH - LT. they only must make some minor modifications as:

In the introduction, it would be adequate to describe the mechanism of ruxolitinib and why the authors chose it! (briefly)
We briefly introduce ruxolitinib in the introduction section (Line 51-55).

Line 46: Briefly report the most common drugs in the HLH-2004 treatment protocol. 

We added the most common drugs in HLH-2004 treatment protocol in Line 46 (now Line 47-48).
Figure 2: bar scale is missing, please put it on the images.

The bar scale was added in figure 2.
Line 172-178, the authors must inform the reference

In Line 172-178 (now Line 208-223) the reference was added.